Fast and accurate semantic annotation of bioassays exploiting a hybrid of machine learning and user confirmation

Clark Alex M. 1 alex@collaborativedrug.com
Bunin Barry A. 1
Litterman Nadia K. 1
Schürer Stephan C. 2
Visser Ubbo 2
1 Collaborative Drug Discovery, Inc. , Burlingame, CA , USA
2 Center for Computational Science, University of Miami , Miami, FL , USA
Kumar Abhishek
Electronic publication date: 2014 Aug 14
Publication date: 2014
Volume: 2
Electronic Location ID: e524
Received 2014 Jul 3; Accepted 2014 Jul 27
Copyright: © 2014 Clark et al.
Copyright year: 2014
Copyright holder: Clark et al.
License: This is an open access article distributed under the terms of the Creative Commons Attribution License, which permits unrestricted use, distribution, reproduction and adaptation in any medium and for any purpose provided that it is properly attributed. For attribution, the original author(s), title, publication source (PeerJ) and either DOI or URL of the article must be cited.
License URL: https://creativecommons.org/licenses/by/4.0/

Keywords: Bioassay, Ontology, Machine learning, Natural language processing, Bayesian, Semantic curation

Funding: NIH RC2 HG005668 U01HL111561 NIH SBIR 1R43TR000185-01A1 The development of BAO and annotation/markup of assays was supported by NIH grants RC2 HG005668 and U01HL111561 to SCS. Modeling and software development was supported by NIH SBIR grant 1R43TR000185-01A1to BAB. The funders had no role in study design, data collection and analysis, decision to publish, or preparation of the manuscript.

==============================
Bioinformatics and computer aided drug design rely on the curation of a large number of protocols for biological assays that measure the ability of potential drugs to achieve a therapeutic effect. These assay protocols are generally published by scientists in the form of plain text, which needs to be more precisely annotated in order to be useful to software methods. We have developed a pragmatic approach to describing assays according to the semantic definitions of the BioAssay Ontology (BAO) project, using a hybrid of machine learning based on natural language processing, and a simplified user interface designed to help scientists curate their data with minimum effort. We have carried out this work based on the premise that pure machine learning is insufficiently accurate, and that expecting scientists to find the time to annotate their protocols manually is unrealistic. By combining these approaches, we have created an effective prototype for which annotation of bioassay text within the domain of the training set can be accomplished very quickly. Well-trained annotations require single-click user approval, while annotations from outside the training set domain can be identified using the search feature of a well-designed user interface, and subsequently used to improve the underlying models. By drastically reducing the time required for scientists to annotate their assays, we can realistically advocate for semantic annotation to become a standard part of the publication process. Once even a small proportion of the public body of bioassay data is marked up, bioinformatics researchers can begin to construct sophisticated and useful searching and analysis algorithms that will provide a diverse and powerful set of tools for drug discovery researchers.

Introduction

In recent decades scientific data has been almost entirely digitized: authors prepare their manuscripts and presentations using a collection of text, graphics and data processing software. Consumers of scientific data regularly download documents from publishers’ websites, search for content in databases, and share data with their colleagues electronically, often in an entirely paperless fashion. Dozens of commercial and academic research groups are actively working on ways to use software to analyze this rapidly expanding corpus of data to provide facile information retrieval, and to build decision support systems to ensure that new research makes the best possible use of all available prior art.

Despite the near complete migration from paper to computers (Khabsa & Giles, 2014), the style in which scientists express their results has barely changed since the dawn of scientific publishing. Whenever possible, ideas and facts are expressed as terse paragraphs of English text, kept as short as possible to minimize printing costs, and as stripped down diagrams that often summarize vast numbers of individual data points in a form that can be visualized statically by a scientific peer. These methods of communication have remained consistent because they are effective for their primary purpose, but this presents a major hurdle to computer software that is attempting to perform data mining operations on published results.

In the case of biological assays, experiments designed to measure the effects of introduced substances for a model of a biological system or disease process, the protocols are typically described in one or more textual paragraphs. Information about the target biology, the proteins or cells, the measurement system, the preparation process, etc., are all described using information rich jargon that allows other scientists to understand the conditions and the purpose. This comprehension process is, however, expert-specific and quite time consuming. While one scientist may read and understand dozens of published assay descriptions, this is not scalable for large-scale analysis, e.g., clustering into groups after generating pairwise metrics, or searching databases for related assays.

One of the most promising approaches to solving this problem is to express the assay design experiments with terminology from a semantically rich ontology, which has the advantage of being readily understood by software (Jonquet, Musen & Shah, 2010; Jonquet, Shah & Musen, 2009; Roeder et al., 2010). Efforts such as the BioAssay Ontology (BAO) project (Abeyruwan et al., 2014; Vempati et al., 2012; Visser et al., 2011) were specifically designed to address this issue, and is part of a pantheon of ontologies for expressing the chemistry and biology definitions and relationships that are essential to drug discovery. Having all relevant scientific data expressed in semantic form enables an inordinate number of options for building compelling decision support software, but the biggest hurdle is the expression of the data. Expecting scientists to alter their documentation habits to use computer-friendly ontologies rather than human-friendly natural language is unrealistic, especially given that the benefits do not start to accrue until a critical mass is achieved within the community. On the other hand, there has been a considerable amount of research toward designing software to perform fully automated parsing of otherwise intractable text and diagrams (Attwood et al., 2010), and add annotations in a machine-friendly format. Many of these efforts have been found to be valuable for certain scenarios where the high error rate is tolerable. For example, allowing a scientist to search the entire patent literature for chemical reactions may be a very useful service even with a low signal to noise ratio, because the effort required to manually filter out false positives is relatively low, and the portion of false negatives may be no worse than more traditional methods (Hawizy et al., 2011; Jessop, Adams & Murray-Rust, 2011; Jessop et al., 2011).

Nonetheless, such fully automated extraction procedures are likely to continue to have a very high error rate for most scientific subject areas for the conceivably near future, and the poor signal to noise ratio prevents most kinds of analysis from being effective. To address this urgent issue, we have developed methods for combining automated extraction and manual curation in order to optimize for both goals: minimal additional burden on practicing scientists, and minimal transcription errors during the semantic markup process.

There are already examples of hybrid manual/automatic annotation technologies, for example PubTator, (Wei, Kao & Lu, 2013) which is designed to help identify a variety of keywords in order to classify papers within the PubMed collection. The web interface provides an initial attempt to identify keywords that correspond to semantic content in the chemical or biological domain, and allows the user to confirm them or add their own. On the other hand, some of the large scale curation efforts, such as ChEMBL, provide funding for expert curators to manually annotate bioassay data, but this is too labor intensive to execute in detail, and is currently limited to identifying the target (Gaulton et al., 2012). Approaches such as Active Learning (AL) have also been applied to classification of domain specific text documents (Cohn, Ghahraman & Jordan, 1996; Dara et al., 2014; Tomanek & Hahn, 2007). Our objective in this work is to provide the necessary capabilities to annotate bioassay protocols, in a significant level of detail, such that the semantic content is a relatively complete description of the assay. In many ways our aims are similar to other natural language text-based classification projects, but unusual in that we are ultimately seeking to use these methods to express a very detailed description of a domain localized class of experiments. We can draw upon existing vocabularies, such as the BioAssay Ontology (BAO), and other ontologies, which it in turn references, for the means to complete this description. To achieve the objective of reducing the burden on the individual scientist to the bare minimum, we have made use of natural language processing and machine learning techniques, and coupled the algorithms to a prototype user interface with a workflow design that iterates back and forth between automated inference and operator approval.

Rather than starting with the lofty objective of having an algorithm provide the right answers all of the time, we merely require it to eliminate most of the wrong answers. To the extent that we are able to achieve this comparatively realistic goal, this allows us to create a user-facing service for which the scientist simply selects correct semantic markup options from a short list of options proposed by the software. This is as opposed to the entirely manual curation approach, which would require the operator to navigate through a densely packed hierarchy of descriptors. By reducing the burden of markup to mere minutes by somebody who is not familiar with semantic technology, and has had no special training for use of the software, it is quite reasonable to expect scientists to use this software as part of their standard write-up and submission process.

As the number of correctly annotated bioassay protocols grows, it will improve the training set and the machine learning algorithm will correspondingly improve in accuracy. Once the currently high barrier to adoption has been overcome, and semantic markup of scientific knowledge such as biological assay experiments is routine, assay protocols will be as readable to computer software as they are to expert scientists. The informatics capabilities that this will unlock are essentially limitless, but the first most clear example is the ability to search assays for specific properties, e.g., target, assay type, cell line, experimental conditions, etc. Being able to conveniently organize and aggregate assays by particular characteristics, cluster by similarity, or assemble chemical structures and activity from multiple assays based on customizable criteria, are all advantages that have a direct impact on drug discovery, which are currently held back by the lack of semantic annotation. Once the corpus of marked up annotations becomes large, it will also be possible to construct data mining algorithms to study large scale trends in bioassay data, which will result in entirely new kinds of insight that are currently not possible.

Methods

Ontologies

The primary annotation reference for this project is the BioAssay Ontology (BAO), which is available from http://bioassayontology.org, and can be downloaded in raw RDF format. The BAO classes refer to a number of other ontologies, and of particular relevance are the Cell Line Ontology (CLO) (Sarntivijai et al., 2011), Gene Ontology (GO) (Balakrishnan et al., 2013; Blake, 2013), and NCBI Taxonomy (Federhen, 2012), all of which are used for annotations within the training set. All of the source files for these ontologies were loaded into a SPARQL server (Apache Fuseki) (The Apache Software Foundation, 2014a). SPARQL queries were used to organize the available values that correspond to each of the property groups.

Training data

In order to test the methodology of using text to create suggested annotations, we made use of a corpus of annotated bioassays that were provided by the BAO group (Schurer et al., 2011; Vempati et al., 2012) (see Supplemental Information). As part of the testing process for the BioAssay Ontology project, a simple annotation user interface was created in the form of an Excel spreadsheet template. Approximately 1,000 assays were selected from PubChem, and each of these was painstakingly annotated, leading to an output document taking the form of: 〈assayID〉 〈property〉 〈value〉.

For each assay, 20–30 annotations were incorporated into the training set. The property values were individually mapped to the BAO space, e.g., ‘has assay method’ is mapped to the URI http://www.bioassayontology.org/bao#BAO_0000212, which is a part of the BAO ontology. Values that are string literals are not included in the training data. Those which map to a distinct URI are typically part of the BioAssay Ontology directly, or part of other ontologies that are referenced, such as the Cell Line Ontology (CLO), Gene Ontology (GO) and NCBI Taxonomy.

Once the annotations had been suitably collated for each distinct assay, the NCBI PubChem assays were obtained by a simple script making a call to the PUG RESTful API (NCBI, 2014). In each case, the description and protocol sections of the resulting content were merged into a free text document. The manner in which these two fields are used by scientists submitting new assay data varies considerably, but they are generally complete. For the collection of text documents obtained, it was necessary to manually examine each entry, and remove non-pertinent information, such as attribution, references and introductory text. The residual text for each case was a description of the assay, including information about the target objective, the experimental details, and the materials used. The volume of text varies from concisely worded single paragraph summaries to verbosely detailed page length accounts of experimental methodology. These reductively curated training documents can be found in the Supplemental Information.

Natural language processing

There has been a considerable amount of effort in the fields of computer science and linguistics to develop ways to classify written English documents in terms of classified tokens that can be partially understood by computer software (Kang & Kayaalp, 2013; Leaman, Islamaj Dogan & Lu, 2013; Liu, Hogan & Crowley, 2011; Santorini, 1990). We made use of the OpenNLP project (The Apache Software Foundation, 2014b), which provides part of speech (POS) tagging capabilities, using the default dictionaries that have been trained on general purpose English text. The POS tags represent each individual word as a token that is further annotated by its type, e.g., the words “report” and “PubChem” were classified as an ordinary noun and a proper noun, respectively:

(NN report) (NNP PubChem)

Blocks of text are classified in an increasingly specific hierarchical form, e.g.,

(NP (DT an) (JJ anti-cancer) (NN drug)) (VP (VBG developing) (NP (JJ potential) (JJ human) (NNS therapeutics))) (NP (NP (NN incubation)) (PP (IN with) (NP (NN test) (NN compound)))) (NP (NP (DT the) (JJ metabolic) (NN activity)) (PP (IN of) (NP (DT a) (NN suspension) (NN cell) (NN line)))) (VP (VB measure) (SBAR (WHADVP (WRB when)) (S (VP (VBG developing)(NP (JJ potential) (JJ human) (NNS therapeutics)))))) (NP (JJ luciferase-based) (NN cell) (NN proliferation/viability) (NN assay) (NN endpoint))

An assay description of several paragraphs can generate many hundred distinct instances of POS-tagged blocks. These marked up tokens contain a far larger amount of information about the composition of the sentence than the words themselves. While building a model by correlating words with annotations would be expected to achieve poor results, including markup information about how the words are used in conjunction with each other might be able to achieve far greater discrimination. For example, the POS-tagged block “(NP (DT an) (JJ anti-cancer) (NN drug))” represents the words [an, anti, cancer, drug]. Each of these 4 words taken out of context could be found in almost any assay description, but when they are associated together in context, contribute an important statement about the corresponding biochemistry.

By collecting all sizes of POS-tagged blocks, up to a certain limit, it is possible to give many different depths of linguistic structure the opportunity to distinguish themselves within a model. In some cases a single word can have significant meaning on its own, especially proper nouns or jargon (e.g., “luciferase”), and are likely to have a high correlation to certain kinds of annotations (e.g., use of a luciferase-based assay). Other words are general to the English language, or occur frequently in assay descriptions, such that they only have value in their proper context (e.g., “interaction”).

One of the useful properties of scientific writing is that authors have self-organized around a narrow range of styles for presenting information such as assay descriptions. While the explicit intent may not have been for the benefit of computerized natural language processing, the motivation is the same: scientific authors also read many other published descriptions, and it is in the best interests of the community to maintain a certain degree of consistency as well as brevity. Because the literary style lacks prose and has a relatively little variation, there are certain blocks of words, as identified by the POS-tagging, that are frequently correlated with particular concepts, and hence the semantic annotations.

Machine learning models

A collection of hundreds of assay descriptions will result in thousands of distinct POS-tagged blocks after processing each of them with natural language analysis, and while certain blocks are likely to be specifically correlated with certain annotations, there are many more with only weak correlation or none at all. Matching thousands of potentially important tags with hundreds or thousands of annotations requires the selection of an algorithm with favorable scaling properties, and is well beyond the scope of manual curation.

In our initial explorations, we chose to apply a variation of Bayesian inference, which has been used successfully in other aspects of computer aided drug discovery. The Laplacian-modified naïve Bayesian variant is frequently used in conjunction with chemical structure based fingerprints (Hassan et al., 2006; Mussa, Mitchell & Glen, 2013; Nidhi et al., 2006; Rogers, Brown & Hahn, 2005), as it is highly tolerant of large numbers of parameters. The score for each annotation is calculated as: score=∑nlnAn+1Tn⋅P+1

where n is the tagged natural language block, An is the number of documents containing the annotation and the tagged block, Tn is the total number of documents with the tagged block, and P is the fraction of documents containing the annotation. The score is computed by adding up the logarithms of these ratios, which circumvents issues with numeric precision, but produces a score with arbitrary scale, rather than a probability.

When we considered each individual annotation as a separate observation, building a Bayesian model using the presence or absence of each distinct POS-tagged block gave rise to a highly favorable response for most annotations, as determined by the receiver–operator-characteristic (ROC) curves. Selected examples of these models are shown in Fig. 1: Fig. 1A shows annotations with high training set coverage that perform well, due in part to having relatively unambiguous word associations, while Fig. 1B shows well covered annotations that perform poorly, due to being reliant on terms that can be used in a variety of contexts that do not necessarily imply the presence of the annotation, and hence make it more difficult for the model to eliminate false positives. Similarly, Fig. 1C shows the perfect recall for less well covered annotations, which are easily identified due to very specific terms, while Fig. 1D shows a relatively poor response due to small training set and terminology with variations in wording style.

Figure 1 Selected leave-one-out ROC plots for annotations, using Bayesian learning models derived from marked-up natural language processing.

One of the disadvantages of using this Laplacian corrected variant is that the computed value is not a probability, but rather a score with arbitrary range and scale. This means that it is not possible to compare the outcomes from two separate models, which is a problem, since the objective of this technology is to rank the scores that are obtained from each separate model. In order to achieve the ranking, the scores need to be directly comparable, and hence be suitable for providing a list of suggestions for which annotations are most likely to be associated with the text.

In order to make the scores from each of the models comparable, each model requires a calibration function. This can be accomplished effectively by defining a simple linear correction for each model, of the form y = ax + b, which is applied to each score prior to inter-model comparison. Selecting appropriate values for a and b, for each model, can be achieved by picking initial values that map each of the model outcomes to the range 0.1. By adjusting the scale and offset of the linear calibration functions for each of these models, the overall ability of the models to correctly rank the extant annotations with a higher score than those which are not observed can be evaluated. It is straightforward to define a scoring term that measures the ability of the calibrated models to distinguish between correct and incorrect annotations. This score can be optimized by iteratively adjusting the calibration terms to get the best overall separation in ranking.

Besides consistent use of linguistic descriptions of assays, one of the other observations about the annotations defined for these assay protocols is that they are not in any way orthogonal: the degree to which the annotations are correlated is very high. For example, if it is known that the assay uses luciferin as a substrate, the probability that it also involves luminescence as a detection method is far higher than it would be if the prior fact had not already been established.

Given that the calibrated Bayesian models have been established to perform very well at placing the top few highest ranking annotations for the data used in this study, once these top scoring annotations have been confirmed by the user, the amount of information that can be inferred about the assay may be significantly greater, due to the high degree of correlation.

This second order correlation was implemented by building another set of Bayesian models with each possible annotation considered separately as an observation. For each document, each annotation’s likely presence is modeled against the presence or absence of all the other annotations recorded for the document, e.g., when building the correlation model for annotation A, if document i contains annotations A, B and C, then it is considered to be “active”, with priors B and C; if document j contains annotations B, C and D, it is considered “inactive”, with priors B, C and D.

Thus, once one or more annotations have been assigned, the secondary Bayesian models are consulted, and the score for each of the annotations is modified by applying the correlation factor. Essentially this means that as the user approves annotations, the prediction scores of the remaining annotations tends to improve, as the correlations are factored in.

Figure 2 provides an indication of how the ranking evolves during the model building steps, using four example documents. For each of these diagrams, the left hand side shows two bands which represent the uncalibrated predictions, which are linearly normalized so their values fall between the minimum and maximum scores from the raw Bayesian prediction score. The annotations that do not apply to the document are shown as red lines, while the annotations that are present are shown in black. The height of each line is indicative of its score. As can be clearly seen, the desired predictions score are significantly higher for those present than those which are absent, but the extent to which the ranking separates the two groups varies, and is not initially a perfect separation for any of these examples.

Figure 2 Representative examples of model building in action, showing relative ranking of uncalibrated, calibrated, and stepwise application of the correlation models.

The four examples refer PubChem entries by assay ID: (A) 574, (B) 436, (C) 348 and (D) 346.

The main area of each diagram shows the progression of the relative predictions: at the beginning of the sequence, the scores are ranked by the inter-model calibration functions, which typically results in a significant improvement. For each of the subsequent steps, the highest scoring correct annotation is added to the approved set, and the correlation model is updated and applied. The ranking is redetermined, and the next highest scoring correct annotation is selected. The diagram indicates the point at which each annotation is selected by plotting a black circle, and changing the color of the line to green: since it has been confirmed, its ranking order is no longer a concern, though its presence continues to affect the way the correlation model is applied.

In the first example, shown in Fig. 2A, application of these models in the given sequence provides a perfect result: in each case the highest scoring annotation yet to be selected is at the top of the list, with no false positives. In Figs. 2B and 2C, the results are good but not perfect: the red cross marks indicate when an incorrect annotation was presented as the best next choice. Since this exercise is simulating curation by a human expert, the elimination of a top-ranked incorrect proposal is equivalent to being recognized by the user as an incorrect result, and explicitly excluded from further consideration. If the objective was to provide a pure machine learning solution, each of these ranking mistakes would represent the accumulation of bad data, rather than a small increase in the amount of effort required by the operator. In Fig. 2D, the response of the model is relatively poor, with several false positives appearing close to the top of the list, and the last few correct results being obscured by a large number of incorrectly ranked proposals.

Results

We have designed the algorithm with the goal of ranking the available annotations such that given a text description of an assay, the annotations that correctly apply to the document are listed before any which do not. A perfect result is considered to be any case where all of the correct proposals are ranked first. Because the objective of the machine learning is to assist and accelerate the human-guided curation, a handful of mis-ordered annotations can be considered as an inconvenience, rather than the means by which data becomes corrupted.

For evaluation purposes, we define a yardstick measure: the null hypothesis is that the Bayesian-trained model using natural language processing performs no better than a trivial method, such as ranking all proposed annotations by the frequency with which they occur in the training set.

Cross validation

The 983 fully annotated assays, with corresponding text from PubChem, were split into training and test sets using a simple partitioning algorithm. First of all, 208 documents were removed on account of having the same list of property:value annotations. These documents may differ by the free text annotations, but these are not included in the training set, and so duplicates need to be pruned. Of the remaining documents, entries were selectively picked for the test set in order to ensure that each annotation appears once in any one of the test set documents, but such that the number of instances remaining in the training set was not reduced below 2. The training set contained 698 documents, the test set 77.

The models were rebuilt using just the training set documents, and applied to the test set. For evaluation purposes, we can consider the ranking of correct vs. incorrect answers to be instructive for how well the model achieves its stated goal. Figure 3 shows several plots that show the relative performance of the training and test sets.

Figure 3 Effectiveness of ranking of activities.

(A) hit/miss for test data; (B) heatmap for model size; (C) null hypothesis; (D) hit/miss for training data.

The data for each plot is created according to the following procedure:

1. score each available annotation based on the model derived from the training set data;

2. pick the highest scoring annotation: if it is correct, add a positive mark, remove the annotation, and goto 1;

3. it is not correct, so add a negative mark, remove the annotation, and goto 2.

This essentially simulates an expert operator who only looks at the current top scoring annotation, and either approves it, or excludes it from further consideration. The process stops when all correct annotations have been approved.

Figure 3 illustrates this process graphically from several vantagepoints. In Fig. 3A, all of the test set documents are considered: for each line, running from left to right, a correct top ranking annotation is marked with a black square, while an incorrect top ranking annotation is marked with a purple square. Once all of the correct annotations have been picked, the remaining space is marked in grey. As can be seen, for the majority of cases the correct annotations are quickly picked out. Nonetheless there are a number of test documents that contain a small number of outliers, i.e., required annotations that are ranked very poorly, with many false positives getting a higher score.

Figure 3B shows the same datapoints, except that only the actual annotations are given a color. The color is determined by a heatmap pattern, for which green indicates predictions that were derived from a well-populated model with many examples, while red indicates those for which very little training data was available. As can be seen, the outliers that rank very poorly relative to the false positives are all colored red, which strongly suggests that poor performance is due to sparsity of training data, rather than flaws with the method.

In Fig. 3C, the method for scoring documents is set to the frequency of each annotation in the overall training set, e.g., if an annotation occurs 100 times in 698 documents, its score is set to 0.143. The same proposed ranking order is used for all documents, regardless of the text description. This is used to test a reasonable null hypothesis, which is that picking the most common annotations is an effective way to separate correct from incorrect. While it can be clearly seen that the null hypothesis performs better than a random guess, at least for purposes of identifying true positives, it is vastly inferior to the proposals generated by the trained Bayesian-derived models, on account of the fact that every document has a very large number of false positives that need to be eliminated before the annotation is complete.

Figure 3D shows the same process as for Fig. 3A, except that in this case the training data is used, i.e., the models are used to predict the same documents from which they were trained. These results are superior to applying the models to the test set, which is to be expected.

Operator workflow

The ultimate goal of combining machine learning with a user interface for bioassay annotation is to have the models predict all the correct annotations with close to perfect accuracy, and have the expert operator confirm these predictions. In practice this is realistic only when the document being annotated consists of cases that are well covered in the training set. Due to the nature of science, there will always be new methods being developed, which means that some of the corresponding annotations may have insufficient examples to create a model. It is also possible that the choice of phrasing for some of the assay details differs significantly from the language used by the examples in the training set, which can reduce the efficacy of the models, until additional data can be incorporated and used to re-train them.

For these reasons, the user interface needs to satisfy two main scenarios: (1) when the predictions are correct, and (2) when the document is unable to accurately predict the annotations. For the first scenario, confirming the correct annotations should be a matter of quickly scanning the highest scoring proposals and confirming the assignments. In these cases, the user interface must aspire to being unobtrusive. However, in the second scenario, when the correct annotation does not appear at the top of the list, the interface needs to provide ways for the user to hunt down the necessary annotations. We have conceived several options to help the user deal with this case. In near-ideal cases, the user may find the correct annotation by simply looking slightly further down the list of proposed annotations. Alternatively, the user may filter the results by selecting a particular category of annotations, and browse through this refined subset to find the correct annotation. Finally, if the user needs to include an annotation that is present in the ontology, but has not been included in the list of proposals because there is not enough data to build a model, the interface can provide assistance in searching through all of the unscored options. Furthermore, there will also be occasions when the desired annotation does not exist in the ontology, e.g., a previously unreported biological pathway, in which case it may be desirable to allow the user to enter the information as plain text. While this has little immediate value for semantic purposes, it could be considered as a placeholder for future additions to the underlying ontology, which could be upgraded retroactively.

A mockup of the core elements of this interface is shown in Fig. 4, which shows the same layout principles for the proof of concept application that we created for testing the machine learning methods and corresponding workflow. The box shown at the top left allows the user to type in free text. This could be cut-and-pasted from another application, or it could be typed in manually. The list immediately below shows a series of annotations, consisting of property and value. These are ranked highest first. When the system is working perfectly, the user can click on the approve button for the highest scoring annotation, shown at the top of the list. If the highest scoring annotation is not correct, the user may look further down the list in order to find one that is correct; or, they may reject an incorrect proposal. In either case, the proposals are recomputed, and a new list of options is shown.

Figure 4 A mockup of an interactive graphical user interface for annotating bioassays, with guidance from pretrained models.

On the right hand side of the screen is shown all of the available properties, which are used to organize the annotations: for each property or category, there can be zero-or-more assigned annotations, of the property:value form. This simple hierarchical arrangement clearly shows the annotations that have been assigned so far, and which properties have as yet no associations. Making the property icons clickable is a way to allow filtering of the annotation list, i.e., only showing the potential annotations that match the selected property. In this way, the operator can carefully pick out assignments for each of the property groups, which is a workflow that becomes important when working with documents that do not fall within the domain of pretrained data. This process of picking out the correct assignment can either be done by scrolling through the list of all possible annotations ranked according to the predictive score, or by partial text searching.

Domain example

Figure 5 shows the annotation of an assay, which can be found in PubChem (Assay ID 761). The annotation text has been composed by concatenating the assay description and protocol text fields, and trimmed to remove superfluous content, which is shown in Fig. 5A. This case is an example where the performance of the machine learning models is strong, but still requires a well-designed user interface for the portions that are less well covered.

Figure 5 Stepwise annotation process for PubChem Assay ID 761.

Steps (B) through (Y) show each of the assignment steps: in most of these examples, the 5 highest ranked annotations are shown. In most of the initial steps, the top ranked case is a correctly predicted annotation. A green checkbox is used to indicate that the user confirms that presence of the annotation, and in the following step, the list of proposals is updated to reflect the modified scores, which take into account the correlation effects. In cases (L), (Q), (R) and (T), the top ranked prediction is incorrect, and a red cross mark indicates that the user explicitly excludes the annotation from further consideration. In step (W) the desired annotation is further down the list, and so the user scrolls the proposals in order to select the next correct one. In step (X), the user needs to add the annotation bioassay type: binding assay, which has not been ranked well in the overall scheme, and so the list of annotations is filtered by selecting the bioassay type property, to only show these corresponding values. In step (Y) the user is looking to find the GPCR signal pathway annotation, which is not a part of the training set, due to insufficient data to build a model. In order to locate this annotation, the user enters a search string to narrow down the list and locate it.

In Fig. 5Z, the complete list of annotations, divided into property categories, is shown. This list is updated dynamically as each of the annotations is added to the collection. The properties for cell line, assay kit and inducer have no corresponding annotations, since these are not a part of the assay.

Semantic output

The purpose of adding semantic annotations to bioassays is to enable a diverse range of queries and automated analysis, and one of the most effective ways to enable this is load the annotation markup into the same framework as the original BioAssay Ontology definition and all of its related dependencies.

The output from an annotated description can easily be expressed in terms of RDF triples. The properties and values are already mapped into the BAO space. A new URI needs to be defined for each of the assays being annotated. For example, the annotation example used earlier, converted into RDF “Turtle” format, is shown in Fig. 6.

Figure 6 RDF Triples for the annotation of PubChem assay ID 761.

Once in this format, the assertions can be loaded into an existing SPARQL server. At this point the content becomes accessible to the full suite of semantic technologies. Combining the generic querying capabilities of the SPARQL syntax, with the semantic structure of the BioAssay ontology, allows a variety of ad hoc questions to be answered.

For example, finding a list of annotated documents that make use of a specific assay kit:

PREFIX rdf: <http://www.w3.org/1999/02/22-rdf-syntax-ns#> PREFIX owl: <http://www.w3.org/2002/07/owl#> PREFIX xsd: <http://www.w3.org/2001/XMLSchema#> PREFIX rdfs: <http://www.w3.org/2000/01/rdf-schema#> PREFIX bao: <http://www.bioassayontology.org/bao#> PREFIX cdd: <http://www.collaborativedrug.com/bao/curation.owl#> SELECT ?aid WHERE { ?assaykit rdfs:label "HTRF cAMP Detection Kit" . ?has rdfs:label "has assay kit" . ?document ?has ?assaykit . ?document cdd:PubChemAID ?aid }

This simple query extracts a list of assay identifiers for anything that makes use of a specific manufacturer’s cyclic AMP detector. Note that the property and value URIs are matched by cross referencing the label. Based on the training data, this query returns AID numbers 933, 940, 1080, 1402, 1403, 1421, 1422 and 488980.

A slightly more advanced query can extract information other than just document identifiers:

SELECT ?instrument ?aid WHERE { ?document bao:BAO_0002855 bao:BAO_0000110 . ?document bao:BAO_0000196 bao:BAO_0000091 . ?document bao:BAO_0000207 bao:BAO_0000363 . ?document bao:BAO_0002865 ?q . ?q rdfs:label ?instrument . ?document cdd:PubChemAID ?aid } ORDER BY ?instrument ?aid

In this case the restrictions are specified by directly referencing the BAO tags, which searches for all protein-small molecule interaction assays, with inhibition as the mode of action, using fluorescence intensity measurements. For each match, the detection instrument is looked up and cross referenced by label:

EnVision Multilabel Reader	622	
PHERAstar Plus	1,986	
ViewLux ultraHTS Microplate Imager	2,323	
ViewLux ultraHTS Microplate Imager	485,281	
ViewLux ultraHTS Microplate Imager	489,008	

The inheritance hierarchy of the BioAssay Ontology, and the ontologies it references, can also be utilized in queries. The following query looks for assays that target GPCRs of mammals:

SELECT ?organism ?aid WHERE { ?mammal rdfs:label "mammalian" . ?target rdfs:subClassOf* ?mammal . ?target rdfs:label ?organism . ?document bao:BAO_0002921 ?target . ?q rdfs:label "G protein coupled receptor" . ?document bao:BAO_0000211 ?q . ?document cdd:PubChemAID ?aid } ORDER BY ?organism ?aid

The target variable is used to match any organism URI that is a subclass of mammals. The result is a number of assays for humans, rats and mice.

Each of these examples shows how the semantic markup of the annotated assays can be put to the test with very direct and specific adhoc questions. These queries can be composed on the fly by software that provides a more intuitive user interface, or they can be used for developing new kinds of analyses by experts. They can be applied to just the bioassay data in isolation, or they can be spliced into the greater semantic web as a whole, and linked to all manner of other information resources, e.g., screening runs measuring drug candidates, or medical knowledgebases that go into more detail about the biological systems being assayed.

Future work

The hybrid interactive/machine learning approach to bioassay annotation is currently a proof of concept product. The prototype user interface is presently being evaluated by scientists with an interest in improving software annotation of biological data, and we are actively assessing the results in order to improve the workflow. The long-term goal is to provide the user interface in the form of a web application, which will be incorporated into larger products that provide data capture functionality, such as the CDD Vault developed by Collaborative Drug Discovery Inc. or potentially public databases such as PubChem. The semantic annotations will be recorded alongside the text description, and immediately accessible, sharable, searchable and used by a variety of features that can provide reasoning capabilities based on this data.

One of the obvious advantages of having user-approved annotations stored in a centralized location is that the machine learning models can be retrained at periodic intervals, which will ensure that the ease with which users can rapidly annotate their assays continues to improve as more data is submitted. Also, as more data becomes available, the domain of the models will continue to grow: annotations that were previously left out of the model building process due to insufficient case studies will be added once they have been used.

Another potential advantage of centralization is to provide a pathway for new semantic annotations, i.e., when the BioAssay Ontology and its dependencies do not provide an appropriate term, users can resort to using a free text placeholder. Such annotations can be examined on a regular basis, and either a manual or automated process can be devised to collect together repeated use of related terms, and define a new annotation (e.g., for a new class of biological target or a new measurement technique). This requires a single authority to decide on a universal resource identifier (URI) for the new term, which could be done by the service provider hosting the data, who may also take the opportunity to retroactively upgrade the existing examples of free text labels to use the freshly minted semantic annotation. We have also demonstrated creating a file containing RDF triples for the resulting annotations for a document, and are looking into harmonizing the data format with the Assay Definition Standard format (ADS/ADF) (de Souza et al., 2014; Lahr, 2014).

In addition to working with potential users of this software, we are also looking to incorporate more public content, from large collection services such as PubChem (Wang et al., 2014; Zhang et al., 2011), BARD (de Souza et al., 2014), ChEMBL (Bellis et al., 2011) and OpenPHACTS (Williams et al., 2012). There are a number of research groups exploring ways to add semantic markup to drug discovery data, including bioassays, and many of these annotations can be mapped to the BAO annotations that we have chosen for this project. Even though we have found in our internal evaluation efforts that annotation time can be plausibly reduced to a matter of minutes, this is still a significant burden to impose on busy scientists, especially if participation is voluntary. As we consider deployment of the service, it is important to ensure that the benefits of assay annotation are realized as early as possible, rather than waiting for critical mass, which might otherwise not be achieved. Allowing scientists to use their annotated assays to easily search for similar assays within a database, or as a convenient way to label and categorize their own collections of assays, are anticipated to be effective strategies to make the technology useful during the early adoption phase.

Thinking broadly, one interesting possible use case for the annotation scheme is to run it in reverse: to have the software use the annotations to assemble a paragraph of plain English text, which is suitable for incorporation into a manuscript. In this case the workflow would likely be quite different, e.g., the user types in a poorly formatted collection of terms involved in the assay in order to help the inference engine rank the likely suggestions, selects the appropriate terms, and then has the text produced by the software. Such a service could be of significant use to scientists who are not experienced with writing assay procedures according to the standard style guides.

As part of our ongoing work, we are evaluating our selection of annotations from the underlying ontology. Our initial prototype is strongly influenced by the training data that we have available, which is the result of hundreds of hours of work by qualified domain experts. We are actively working with biologists and medicinal chemists to determine which properties are of primary importance, and which are secondary, and to expand our collection of training data to reflect the priorities of active drug discovery researchers.

Beyond the use of bioassays and BAO annotations for training data, the methodology developed is broadly applicable and not specific to this domain. We anticipate that there are a number of other distinct subject areas of scientific publications that would be amenable to this treatment, e.g., experimental details of chemical reactions, computational chemistry protocols, and other types of biological protocols beyond drug discovery, such as stem cell differentiation.

Conclusion

We have built a proof of concept framework that involves using machine learning based on plain text assay descriptions and curated semantic markup, and matched this with a user interface that is optimized for making machine-assisted annotation very rapid and convenient when applied to text input that is well within the domain, and moderately efficient for annotating assays that fall outside of the training set. By optimizing both the machine learning and user-centric workflow at the same time, we avoid falling into the traps of both extremes, because both parts complement each other. Annotation of plain text by purely automated methods has been limited by the need to obtain an unrealistic level of accuracy, while purely manual annotation has to overcome a very high motivational barrier, given that most scientists are too busy to take on additional burdens, without an immediate benefit. By establishing that adding a very modest amount of human effort to a well designed automated parser can achieve highly accurate results, we believe that we can make a strong case for the use of this technology in the hands of practicing scientists.

As the quantity of semantically rich annotated data increases, the opportunities for delivering value to scientists increases in tandem. Making annotation easy is the first step, but it needs to be followed by new capabilities. For example, the creators of assay screens should be able to easily compare their experiments with others contained within the knowledgebase, and obtain a list of experiments and published papers with common features. Researchers performing drug discovery modeling studies should be able to gather together compounds that have been screened under certain conditions, and use the annotations to make a judgment call as to whether the measured activities can be incorporated into the same model. Additionally, researchers can search for artifacts, such as compounds that are disproportionately active in luminescent assays. New biological activities may also become mineable; for example, common hits between cell-based assays and target based assays may reveal unknown molecular mechanisms.

Beyond the specific domain of bioassay annotation, we believe that the hybrid approach to high level markup is appropriate to many different areas of science, where use of English text jargon or anachronistic diagrams is the norm for conveying concepts that are amenable to a highly structured description. The understandable reluctance of scientists to redesign their communication methods for the benefits of software, and the inability of software to provide substantially useful results without such richly marked up data, is a proverbial chicken vs. egg scenario that can be observed throughout the scientific disciplines. Combining machine learning with modest demands on scientists’ time, and rapid iteration of improved functionality, is a viable strategy for advancing the goals of computer assisted decision support.

Supplemental Information

Supplemental Information 1 Data used to create the training set

Data used to create the training set, in RDF/TTL format (zipped). Includes the curated text for the assays, and each of the triples used to annotate them.

Click here for additional data file.

Additional Information and Declarations

Competing Interests

Author Contributions

Alex M. Clark, Barry A. Bunin, and Nadia K. Litterman are employees of Collaborative Drug Discovery, Inc.

Alex M. Clark conceived and designed the experiments, performed the experiments, analyzed the data, contributed reagents/materials/analysis tools, wrote the paper, prepared figures and/or tables, reviewed drafts of the paper.

Barry A. Bunin conceived and designed the experiments, reviewed drafts of the paper.

Nadia K. Litterman analyzed the data, contributed reagents/materials/analysis tools, wrote the paper, reviewed drafts of the paper.

Stephan C. Schürer and Ubbo Visser contributed reagents/materials/analysis tools, reviewed drafts of the paper.

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
