# Peer review of "Fast and accurate semantic annotation of bioassays exploiting a hybrid of machine learning and user confirmation"

_PeerJ, doi:10.7717/peerj.524_

## Round 0.1 · original submission · Major Revisions

· Academic Editor

Major Revisions

I see strength of this study & hence I recommend to revise this MS according to questions raised by the two reviewers.

·

Basic reporting

The manuscript is sound and easily readable.

The authors nicely identify how their study will benefit the field of biological curation. The introduction clearly defines the purpose of the work, which is relevant and meaningful to the interoperability of data sets.

Several sentences are problematic in regards to clarity:

Abstact
Line 30 -33 “Well-trained annotations require single-click user approval, while annotations from outside the training set domain can be identified using the search feature of a well-designed user interface, and subsequently used to improve the underlying models.”

The reader has been informed that it is a machine learning approach but does not yet understand the details of the methodologies used, making this sentence difficult to understand.

Line 30 “well-trained annotations” ??

In the Introduction:
Lines 56-64 This paragraph should be made clearer or eliminated while incorporating the thought into next section or elsewhere. The point appears to be that presently humans must read assay protocols and that the manual evaluation is not adapted to large scale analysis. Specific difficulties : Line 61 This comprehension process is, however, expert-specific and quite time consuming, and Line 62… scientist may read and understand dozens of published assay descriptions, this is not scalable for large-scale analysis, e.g. clustering into groups after generating pairwise metrics, or searching databases for related assays.

Line 78 “intractable text and diagrams, “
An appropriate reference for this statement, such as,
“T. K. Attwood, D. B. Kell, P. Mcdermott, J. Marsh, S. R. Pettifer, and D. Thorne. Utopia Documents: linking scholarly literature with research data. Bioinformatics, 26:i540-i546, Sep 2010.”
would be desirable.

Experimental design

In regards to the experimental design, the authors effectively define the boundaries of their efforts, which allows them to make the appropriate conclusions. They clearly describe the limits of their methods and their techniques used to deal with the limits. They present both positive and negative results to offer an unbiased view of their methods/tool.

It is not clear if the software is/will be made available via Open Source. A comment should be added to this effect. There is some mention that the web interface might be incorporated into other platforms but more specific mentioning of the nature of the licensing would be appreciated.

Line 227 : When we considered each individual annotation as a separate observable
“as a separate observation” ?

Validity of the findings

Overall the discussion of the findings is well thought out and correctly presented.

Minor comments:
The discussion of the “Semantic output” should be reduced to just description of the Figure 6 RDF and a reference to where this RDF output can be accessed. The inclusion of the SPARQL queries is a distraction that does not seem to add to the overall quality of the manuscript and should be moved to supplemental material. Also, there is no URL given for the SPARQL endpoint.

In Figure 6, the mode of action has 2 values, inhibition and activation. It would be helpful if there was an explanation about how 2 apparently contradicting results can be reconciled or interpreted.

Additional comments

In Figure 5, the right hand panel (z) probably should be identified as such. The (z) is very small and hard to see.

Reviewer 2 ·

Basic reporting

I am very surprised to see that the article doesn't mention Active Learning (AL). AL is not particularly new, for example applying it to corpus construction in a very closely related field, named entity recognition, was not only proposed but actually demonstrated by Katrin Tomanek in her PhD thesis and attendant publications. Here's an early one from 2007: http://anthology.aclweb.org//D/D07/D07-1051.pdf

The "Machine learning models" section should at the very least make it clear how the approach differs from that set out in Cohn et al. (https://www.jair.org/media/295/live-295-1554-jair.pdf) in 1997, who introduced active learning, and the various approaches that Katrin found in a survey of people doing just this back in 2009: http://www.julielab.de/Staff/Alumni/Dr_+Katrin+Tomanek/Active+Learning+Survey.html

Here's another application, just off the top of the ACL Anthology, to machine translation: http://www.aclweb.org/anthology/E/E14/E14-4036.pdf Again this is a case where people have applied it with actual annotators rather than merely a proof-of-concept.

In short, very similar work, with actual human annotators, is being done all the time, and the article needs to indicate how it fits in with all of this much more clearly.

Experimental design

This is broadly sound. The main concern I have is that it's not explicitly stated what a "fully annotated" assay is. It may simply be that list of annotations in Figure 5, but I am reading between the lines here.

Validity of the findings

No comments.

---

## Round 0.2 · accepted · Accept

· Academic Editor

Accept

Thank you for your submission to PeerJ. Your manuscript is Accepted for publication.